# Due Diligence of a Diastolic Index as a Prognostic Factor in Heart Failure with Preserved Ejection Fraction

**DOI:** 10.3390/jcm12206692

**Published:** 2023-10-23

**Authors:** Shiro Hoshida

**Affiliations:** Department of Cardiovascular Medicine, Yao Municipal Hospital, 1-3-1 Ryuge-cho, Osaka 581-0069, Japan; shiro.hoshida@hosp-yao.osaka.jp; Tel.: +81-72-922-0881; Fax: +81-72-924-4820

**Keywords:** arterial elastance, diastolic elastance, HFpEF, prognosis, sex

## Abstract

Of the existing non-invasive diastolic indices, none consider arterial load. This article reveals points of caution for determining the diastolic prognostic index using a novel index of vascular resistance-integrated diastolic function in old, real-world patients with heart failure with preserved ejection fraction (HFpEF) in Japan. This index represents the ratio of left ventricular diastolic elastance (Ed) to arterial elastance (Ea), where Ed/Ea = (E/e′)/(0.9 × systolic blood pressure), showing a relative ratio of left atrial filling pressure to left ventricular end-systolic pressure. The role of hemodynamic prognostic factors related to diastolic function, such as Ed/Ea, may differ according to the clinical endpoint, follow-up duration, and sex. In HFpEF patients with heterogenous cardiac structure and function, an assessment using a serial echocardiographic diastolic index in clinical care can provide an accurate prognosis.

## 1. Introduction

The circadian pattern of blood pressure is mainly affected by autonomic nerve activity. Changes that occur throughout the day affect the cardiovascular system, possibly leading to various alterations in cardiac function, including diastolic function, based on the time and circumstances. Doppler echocardiography is the most useful tool for routinely measuring diastolic function noninvasively. According to the recommendations for left ventricular (LV) diastolic evaluation using echocardiography, the severity of diastolic dysfunction is assessed using a combination of several indices for the left and right sides of the heart, including E/A, deceleration time, E/e′, tricuspid regurgitation velocity, left atrial (LA) strain pattern, and LA volume [1,2,3]. The evaluation of disease severity is useful for prognostic evaluation in patients with heart failure (HF) with preserved ejection fraction (HFpEF) [4]. However, there is important crosstalk between arterial load and diastolic function [5]: diastolic relaxation is reduced with acute increases in the arterial load [6,7,8]. Unfortunately, none of the noninvasive diastolic indices consider systemic vascular resistance. This article reveals the effective modulators for the significance of diastolic prognostic factors through the usefulness of a novel vascular resistance-integrated diastolic index in real-world Japanese patients with HFpEF.

## 2. Validity of a Vascular Resistance-Integrated Diastolic Index

Historically, HFpEF has been thought to be driven by increased afterload in arterial stiffness and hypertension, with consequential LV diastolic dysfunction as one of the hallmarks of the syndrome. Early studies suggested that, in the clinical setting, E/e′ is a marker of reduced LV long-axis early diastolic expansion, that is, one of the markers of LV diastolic function, resulting in an estimated LA filling pressure [9,10]. The correlation between E/e′ and direct LA pressure or pulmonary capillary wedge pressure is significant in a stable state [11,12]. Among several indices evaluated using Doppler echocardiography, E/e′-related indices, such as (E/e′)/stroke volume (SV), i.e., operant diastolic elastance (Ed), reportedly reflect LV diastolic function [13,14]. The effective arterial elastance (Ea) is calculated as (0.9 × systolic blood pressure)/SV [13]. We previously reported on age- and sex-related differences in LV diastolic function relative to arterial elasticity among hypertensive patients with preserved LV ejection fraction (LVEF) and no history of HF [15]. We found that the vascular resistance-integrated diastolic index Ed/Ea (calculated as (E/e′)/(0.9 × systolic blood pressure)) was significantly increased in older (aged ≥ 75 years) hypertensive women and was coincident with cardiac structural alterations. Under stable conditions, Ed and Ea are higher in women than in men [16,17]. Since LV diastolic function is strongly affected by arterial load, higher Ed in women is partly related to an associated increase in Ea. In this sense, it would be better to correct Ed, considered together as Ed/Ea, when analyzing sex differences in diastolic function (Table 1). E values are significantly higher and mean e′ values are significantly lower in HFpEF patients showing high Ed/Ea levels than in those showing low Ed/Ea levels [18]. Although systolic pressure might decrease as SV decreases, there are no significant differences in LV volume and SV between patients with high and low Ed/Ea under the preserved LVEF before discharge [18]. A high ratio of Ed/Ea is not just a sign of reductions in LV volume and filling.

Increased LV filling pressure owing to exercise correlates with changes in the diastolic relaxation rate and arterial afterload [20,21]. The linear slope of the single-beat diastolic pressure–volume relationship is defined as Ed [22,23]. Exercise induces an increase in Ed, evaluated invasively [20] and noninvasively ([E/e′]/SV) [24]. In individual patients, the ability of echocardiography to track changes in left-side filling pressure induced by volume change [25] or exercise [12] is controversial. In previous studies, changes in the arterial load were not considered during the assessment. Arterial load could be assessed using effective arterial elastance, Ea, (Ea = end-systolic pressure/SV) [13,26]. It is essential to measure systolic blood pressure around the examination of echocardiography to measure this index. Advanced age and female sex are associated with increases in arterial and ventricular stiffness, even in the absence of cardiovascular disease [13]. The changes in Ea induced by exercise are different between patients with HFpEF and the controls [21]. According to the study by Borlaug et al. [20], Ed/Ea would not change significantly after stress if it were not for stress-induced ischemia, although the mechanisms underlying the autonomic cardiovascular response to exercise are complicated [27]. Changes in Ea, in addition to those in diastolic elastance, are compromised in HFpEF [28]; these changes are outside of those associated with aging or hypertension [29]. 

We recently reported that the LA volume index and Ed/Ea are high in patients with HFpEF [30]. Ed/Ea reflects the ratio of LA filling pressure to LV end-systolic pressure [31], which means relative LA pressure to LV pressure, which may change minimally over a day under various circumstances. Thus, the Ed/Ea ratio could reflect the left-sided heart function status, including the atrioventricular–arterial interaction, under preserved LVEF conditions. Although a quantitative assessment of the LA conduit function is reported to be a new parameter for LV diastolic dysfunction in HF patients [32], its role in the prognosis of patients with HFpEF remains undefined.

## 3. Utility of the Vascular Resistance-Integrated Diastolic Index for Prognosis in HFpEF

The pathophysiology of HFpEF is complex and includes alterations in cardiac structure and function, systemic and pulmonary vascular abnormalities, and comorbidities [33]. HFpEF is detectable in a relevant portion of antiphospholipid syndromes [34]. Hospitalizations related to HFpEF are increasing, and the growing older population has led to this trend worsening. We recently reported prognostic data in older patients (mean age: 81 years) recruited from the Prospective Multicenter Observational Study of Patients with Heart Failure with Preserved Ejection Fraction registry [35], and found that Ed/Ea is a useful prognostic marker in patients with HFpEF [18,36]. Our findings can help determine which single index of LV diastolic function is significantly associated with the prognosis. In particular, in patients with a higher level of N-terminal pro-brain natriuretic peptide (NT-proBNP), a higher Ed/Ea was associated with a poor prognosis [36]. In a recent prospective study among patients with established HFpEF, Ea [37] and Ed [36] failed to predict adverse outcomes. The other indices for arterial stiffness, carotid–femoral pulse wave velocity (cf-PWV) and the carotid augmentation index (cAIx), are well established [38,39,40]. However, Huang et al. reported that cf-PWV, but not cAIx, was related to clinical outcomes in patients with HFpEF [40]. The patient characteristics were quite different between Huang’s study and our study; the incidences of male gender and coronary artery disease were high in their study, and those of hypertension and atrial fibrillation (AF) were low [40,41]. Furthermore, the most noticeable aspect is that important clinical variables were not included in their analysis of a multivariable model [40]. Although arterial stiffness itself may be of added prognostic value in studies with a longer follow-up [40], cf-PWV was not reported as a predictor for all-cause mortality in patients with HFpEF [38,39,40]. Our proposed index may be easily applicable in hospitals and clinics because the precise measurement of volumetry or strain pattern for the cardiac chambers via echocardiography is not needed.

## 4. Factors Affecting the Significance of Ed/Ea in the Prognosis of Patients with HFpEF 

The prognostic factors related to LA overload may differ according to the clinical endpoint in older patients with HFpEF [41]. Ed/Ea and SV/LA volume (LAV), a relative index for LA volume overload [30], were significant prognostic factors for readmission in HF [40]. However, Ed/Ea, but not SV/LAV, was a significant prognostic factor for all-cause mortality. E/e′ was reported to be a validated predictor for HF readmission but not all-cause mortality in a univariable model after the first acute HFpEF event [42,43]. When E/e′ was used in place of Ed/Ea in our study, E/e′ was not a prognostic factor for readmission for HF in a multivariable Cox hazard analysis [41]. The cutoff point of Ed/Ea for all-cause mortality (nearly 0.130) or readmission for HF (nearly 0.100), observed in a receiver operating characteristic curve analysis during a short-term period in patients with HFpEF [41], was in accordance with that in hypertensive patients with preserved LVEF without HF [15] (mean ± SD value of Ed/Ea, 0.100 ± 0.030; mean age: 80 years; unpublished observation). In older patients with HFpEF, the most important events are readmission for HF in each subject and socioeconomic status. Recently, average and healthy life expectancy has increased all over the world, and the evaluation of prognosis for mortality is more important than ever before within super-aging societies and declining populations, like in Japan.

The prognostic significance of Ed/Ea may only be valid over a short-term period in older patients. Ed/Ea assessed before discharge was a significant prognostic factor for all-cause mortality during the first, but not the second, year after discharge [35]. In landmark Kaplan–Meier survival and multivariable Cox hazard analyses performed using the value of Ed/Ea at 1 year after discharge, Ed/Ea remained a significant prognostic factor during the second year after adjusting for age and sex [18]. To strictly evaluate the prognostic risk for all-cause mortality, serial examination for Ed/Ea would be optimal in a clinical setting. No differences in LVEF or LV mass index (LVMI) were observed between patients with and without events during the first year among those with a high Ed/Ea before discharge; however, a reduced LVEF and larger LVMI were observed in patients with events during the second year among those with high Ed/Ea at 1 year after discharge [18]. The cause of death or the pathophysiology of HFpEF may differ in the first and second years after discharge. The clinical significance of prognostic factors related to hemodynamics in patients with HFpEF may differ based on the follow-up period. The serial measurement of Ed/Ea is needed to assess the occasional prognosis accurately. Under these conditions, one can change a patient’s medications as needed while observing their pathophysiology. In this sense, the role of LVEF in prognosis may be the same as that of Ed/Ea [44,45]. A previous study reported that the most recent Kansas City Cardiomyopathy Questionnaire score was most strongly associated with subsequent prognosis in serial health status evaluations of patients with HFpEF [46]; this corresponds with our findings. 

Arterial stiffness may preferentially contribute to abnormal diastolic function during exercise in women with HFpEF compared with men [47]. Originally, Ed/Ea was used as an index for the sex-related differences in cardiac function in older hypertensive patients with preserved LVEF [15]; however, the sex differences in Ed/Ea as a prognostic indicator remained undefined. We recently reported that between-sex differences in the significance of Ed/Ea as a prognostic factor were observed in patients with HFpEF [48]. Ed/Ea was significant for readmission for HF during the first year in men, but not in women, in a multivariable analysis [48]. P for the interaction regarding Ed/Ea for this prognosis between sexes was significant. In contrast, Ed/Ea was significant for all-cause mortality during the first year after discharge in women, but not in men. Although Ed/Ea was a significant prognostic factor for all-cause mortality in HFpEF patients, including both sexes [41] or each sex (unpublished observation), during the three years after discharge, the majority of the differences in the prognosis for all-cause mortality between the patients with high and low Ed/Ea were observed during the first year [18]. During the short-term period after enrollment, the role of a hemodynamic index such as Ed/Ea as a prognostic factor may be different between sexes. Effective modulators for hemodynamic diastolic indices such as Ed/Ea as a prognostic factor in patients with HFpEF are shown in Table 2.

The significance of Ed/Ea as a prognostic risk factor may be reduced in HFpEF patients with AF because the calculation of Ed/Ea is based on the correct measurement of E/e′, which should be carefully and precisely measured. The incidence of AF did not differ significantly in our patients (38%) [48] compared to the previous report (40%) [49]. E/e′ is correlated with invasive LV filling pressure and adequate reproducibility, even in patients with AF [50]. The R-R interval is irregular in AF, and we measured the mean value of E/e′ among several beats in AF patients with an unstable blood pressure. However, changes in E/e′ may parallel those in blood pressure, and the ratio of E/e′ to blood pressure, i.e., Ed/Ea, does not differ largely under stable conditions. In fact, Ed/Ea did not differ significantly between HFpEF patients with and without AF before discharge [51]. Although Ed/Ea may provide less important information for evaluating all-cause mortality in HFpEF patients with AF than with sinus rhythm [51], in a larger number of HFpEF patients, with or without AF, Ed/Ea is a significant prognostic factor of all-cause mortality [41]. Ed/Ea exhibits the relative and not the absolute value of LA filling pressure and could be representative of the general performance of the left-sided heart under preserved LVEF, including patients with AF. The incidence of AF, but not the Ed/Ea level, is no longer a significant prognostic factor for all-cause mortality in a multivariable model including NT-proBNP [41]. The measurement of systolic blood pressure and E/e′ via echocardiography may be the first-line clinical procedure in HFpEF patients with a history of hospitalization, irrespective of the incidence of AF, in addition to the evaluation of laboratory data such as NT-proBNP to predict their prognoses.

## 5. Futured Perspectives

Because all of the study populations using our proposed index for prognostication included those with hospitalization only, this limits the generalizability of our findings to HFpEF outpatients with no history of hospitalization. The possible clinical utility of this unique index is expected to reveal older, symptomatic outpatients with HFpEF and “pre-HFpEF” outpatients without symptoms [19,52]. The criticism is that there is no external validity to this index. Most of our cited research is derived from a single study, the PURSUIT HFpEF, involving collaborating hospitals in the Osaka region of Japan. Further, in a series of our studies from the PURSUIT registry, the Ed/Ea in HFpEF was not compared to that in controls in the same registry. It is unknown whether this index might be important for establishing prognoses in other HFpEF cohorts. Phenotypic differences may exist in HFpEF patients between Japan and other geographical regions. Therefore, careful characterization is needed to detect new therapies specific to the Japanese population [53]. Since the age of the subjects included in major papers [54,55] showing established results regarding the pharmacological interventions in patients with HFpEF was much younger than that in our study subjects [41], large-scale prospective studies are required to investigate the age-, sex-, follow-up duration-, and geographical region-related differences in the clinical significance of Ed/Ea for the prognosis of HFpEF patients with and without hospitalization.

## 6. Limitations

The assessment of arterial load and ventricular–vascular coupling provides important physiological information and is increasingly used in clinical and epidemiological human research. However, the limitations of this study must be considered. The currently available data are limited, which limits our conclusions. Aside from sex differences found in non-HF patients showing natural differences between sexes in the Ed/Ea index [15], the scarcity of data could also explain why the results are relevant for mortality prognostication in some data but not in others. The precise reasons for the disparity in the prognostic value of Ed/Ea between sexes remain undefined. There is a problem with the definition of Ed and Ea, and the practical implications, such as therapeutic or follow-up strategies, for this index remain undefined. It is difficult to mention Ed as strictly being a “linear” slope of the diastolic pressure–volume relationship, but it is known to be “exponential”. Arterial load is influenced by arterial health and is a key determinant of LV systolic and diastolic function and LV remodeling [56]. Although Ea is attractive because of its simplicity, it has some disadvantages as an index of arterial load. A basic set of parameters that characterizes arterial load includes systemic vascular resistance (SVR) and pulsatile loads [57,58]. The notion [59] that Ea represents a lumped parameter of arterial load incorporating both resistive and pulsatile load is not always acknowledged. Ea may simply be a function of SVR and heart rate that is negligibly influenced by changes in pulsatile afterload in humans [60]. Therefore, I present Ed/Ea as a vascular resistance-integrated index for diastolic function in this text. Even if Ea represents the ratio of mean aortic pressure (MAP) to SV [61], the ratio of (0.9 × SBP)/SV indicates no real change. The correlation between MAP/SV and (0.9 × SBP)/SV was quasi-perfect in patients with HFpEF (r = 0.975, n = 757, unpublished observation). 

## 7. Conclusions

Our novel index of vascular resistance-integrated diastolic function, Ed/Ea = (E/e′)/(0.9 × systolic blood pressure), was valid and useful for evaluating the prognosis of elderly patients with HFpEF. However, factors such as the clinical endpoint, the follow-up duration, and sex may influence the prognostic significance of Ed/Ea. In clinical care, a serial noninvasive index such as Ed/Ea can provide an accurate assessment of the occasional prognosis of patients with HFpEF.

## 8. Supplementary Information

Prognostic findings were obtained from the PURSUIT HFpEF (Prospective, Multicenter, Observational Study of Patients with Heart Failure with Preserved Ejection Fraction) registry [35]. The PURSUIT HFpEF registry is a prospective, multicenter observational study in which collaborating hospitals in the Osaka region of Japan recorded the clinical, echocardiographic, and outcome data of patients with HFpEF (UMIN-CTR ID: UMIN000021831). This study complied with the tenets of the Declaration of Helsinki, and the protocol (Osaka University Clinical Research Review Committee, R000024414) was approved by the ethics committee of each participating hospital (Ex. Ethics Committee of Yao Municipal Hospital, 2016-No.0006). Briefly, hospitalized patients with HF and an LVEF of ≥50% were prospectively registered. All methods were performed in accordance with the relevant guidelines and regulations. All patients provided written, informed consent to participate. We excluded patients with considerable mitral or aortic valve disease.

## Figures and Tables

**Table 1 jcm-12-06692-t001:** A novel index of diastolic function taking vascular resistance into account.

** *Existing index:* **	
Operant diastolic elastance: **Ed** = (E/e′)/stroke volume	
Effective arterial elastance: **Ea** = (0.9 × systolic blood pressure)/stroke volume	
(Redfield M et al. Ref. [13])	
** *Novel index:* **	
Vascular resistance-integrated diastolic index:	
	**Ed**/**Ea**
	= (E/e′)/(0.9 × systolic blood pressure)
	≈ left atrial filling pressure/left ventricular end-systolic pressure
(Hoshida S et al. Refs. [15,19])	

**Table 2 jcm-12-06692-t002:** Reported modulators for a diastolic index, Ed/Ea, as a prognostic factor in heart failure with preserved ejection fraction.

Follow-up duration	(Ref. [18])
Natriuretic peptide level	(Ref. [36])
Prognostic endpoint	(Ref. [41])
Sex	(Ref. [48])

## Data Availability

The datasets used and/or analyzed during the current study are available from the corresponding author on reasonable request.

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
