# Peer review of "Due Diligence of a Diastolic Index as a Prognostic Factor in Heart Failure with Preserved Ejection Fraction"

_jcm, 2023, doi:10.3390/jcm12206692_

Round 1
Reviewer 1 Report
Current manuscript is average.
Very attractive the title and good analysis.
Nevertheless, there are also some POINTs of WEAKNESSES:
-English language to be checked. Sentences such as at lines 9-14 might be totally re-written
-Quality of table 1 to be clarified.
-References that are not always in accordance with authors' rules.
Finally, I'd like to suggest the following articles in order to ameliorate paper and/or insert in references.
Am J Physiol Heart Circ Physiol. 2015 Nov;309(9):H1392-406. doi: 10.1152/ajpheart.00436.2015.
Clin Res Cardiol. 2016 Jan;105(1):17-28. doi: 10.1007/s00392-015-0882-8.
English language to be checked. Sentences such as at lines 9-14 might be totally re-written.
Author Response
Reviewer #1
-English language to be checked. Sentences such as at lines 9-14 might be totally re-written
I checked English language throughout the text (via MDPI). I rewrote the Abstract section according to your suggestion: “This article reveals the cautious points for determining the diastolic prognostic index using a novel index of vascular resistance-integrated diastolic function in old, real-world patients with heart failure with preserved ejection fraction (HFpEF) in Japan. This index represents a ratio of left ventricular diastolic elastance (Ed) to arterial elastance (Ea), where Ed/Ea = (E/e´)/(0.9 × systolic blood pressure), showing a relative ratio of left atrial filling pressure to left ventricular end-systolic pressure.” (page 2, lines 3-9).
-Quality of table 1 to be clarified.
Thank you for your indication. I added a new Table 1 in addition to the corrected Table 2 (old Table 1). Table 2 shows the modulators for Ed/Ea as a prognostic index previously reported.
-References that are not always in accordance with authors' rules.
Since there is a relative high proportion of our papers in all references, I omitted two our papers in the revised manuscript (old refs. 16 and 34).
Finally, I'd like to suggest the following articles in order to ameliorate paper and/or insert in references.
I added two important papers in the revised manuscript according to your suggestions (Refs. 26 and 31).

Reviewer 2 Report
The topic highlighted in the manuscript is interesting. I believe this might be
published after some minor corrections.
I would recommend
-to change the title by using more common words;
-add the scheme explaining a new index;
-add the table summarizing all existing non-invasive parameters used for similar purposes.
Minor corrections are required
Author Response
Reviewer #2
I would recommend
-to change the title by using more common words;
Thank you for your comments. According to English editing by MDPI Susy, the title of my manuscript is not changed.
-add the scheme explaining a new index;]
I added a new Table 1 explaining a novel index.
-add the table summarizing all existing non-invasive parameters used for similar purposes.
Thank you for your advice. I narrowed down the number of effective modulators for Ed/Ea as a prognostic factor in patients with HFpEF in Table 2 (old Table 1), because the modulators for Ed/Ea as a prognostic factor are limited.

Reviewer 3 Report
The authors of the article would like to draw attention to the fact that none of the existing non-invasive diastolic indices consider arterial load. This article reveals the cautionary points for the determination of the diastolic prognostic index through the usefulness of a new index of diastolic function integrated with vascular resistances, a relationship between left ventricular diastolic elastance (Ed) and arterial elastance (Ea), Ed/Ea = (E/e´)/(0.9 × systolic arterial pressure), in the prognostic evaluation of elderly real-world heart failure patients with preserved ejection fraction (HFpEF) in Japan. The role of haemodynamic prognostic factors related to diastolic function, such as Ed/Ea, may vary depending on the clinical endpoint, duration of follow-up and gender. In HFpEF patients with heterogeneous cardiac structure and function, assessment by serial echocardiographic diastolic index in clinical care may provide an accurate prognosis.
A new index of diastolic function integrated with vascular resistance, Ed/Ea = (E/e´)/(0.9 × systolic blood pressure), was found to be valid and useful for assessing the prognosis of elderly patients with HFpEF. However, factors such as clinical endpoint, duration of follow-up and gender may influence the prognostic significance of Ed/Ea. In the clinical setting, a non-invasive serial index such as the Ed/Ea can provide an accurate assessment of the occasional prognosis of patients with HFpEF.
The article is well-written and fairly novel in that it adds an important parameter in the assessment of the prognosis of HFpEF.
I recommend a review of the scientific English
I recommend a review of the scientific English and the need to use shorter sentences that are better understood by the reading public
Author Response
Reviewer #3
I recommend a review of the scientific English and the need to use shorter sentences that are better understood by the reading public.
I checked English language throughout the text. I rewrote the Abstract section to use shorter sentences (page 2, lines 3-9).

Reviewer 4 Report
1. HFpEF is a very pressing problem today. Despite existing guidelines, its diagnosis and management remain puzzling. In this regard, the presentation of the novel index related to the pathophysiological mechanisms of HFpEF seems important and appropriate. One of the main mechanisms is abnormal coupling between the left ventricle and the systemic arterial tree. The novel index of vascular resistance-integrated diastolic function index has a simple formula. There are no questions about its first component: e/e` as an indicator of diastolic function has been well studied and is therefore convincing. As for arterial elastance, its definition seems to be not entirely accurate and studied. Therefore, further research is needed.
2. I do not agree with the author's statement in the "Introduction" (lines 28,29). The “ventricular” strain is not an indicator of diastolic function and should be removed from approaches to assessing diastolic disfunction, since it reflects systolic function - the deformation of the myocardium during contraction (systole).
In general, in spite of these minor comments, the author's point of view on this novel approach to assessing the ventricular-arterial state in HFpEF may be recommended for publication.
Author Response
Reviewer #4
- HFpEF is a very pressing problem today. Despite existing guidelines, its diagnosis and management remain puzzling. In this regard, the presentation of the novel index related to the pathophysiological mechanisms of HFpEF seems important and appropriate. One of the main mechanisms is abnormal coupling between the left ventricle and the systemic arterial tree. The novel index of vascular resistance-integrated diastolic function index has a simple formula. There are no questions about its first component: e/e` as an indicator of diastolic function has been well studied and is therefore convincing. As for arterial elastance, its definition seems to be not entirely accurate and studied. Therefore, further research is needed.
I agree with your opinion. Therefore, I added several sentences in the Limitation section regarding the definition of Ea in the revised manuscript (page 10, lines 18-24).
- I do not agree with the author's statement in the "Introduction" (lines 28,29). The “ventricular” strain is not an indicator of diastolic function and should be removed from approaches to assessing diastolic disfunction, since it reflects systolic function - the deformation of the myocardium during contraction (systole).
Thank you for your advice. I rewrote the Introduction section in the revised manuscript according to your advice (page 2, line 23 – page 3, first line).
